# Identification of the Potential Molecular Mechanisms Linking RUNX1 Activity with Nonalcoholic Fatty Liver Disease, by Means of Systems Biology

**DOI:** 10.3390/biomedicines10061315

**Published:** 2022-06-03

**Authors:** Laia Bertran, Ailende Eigbefoh-Addeh, Marta Portillo-Carrasquer, Andrea Barrientos-Riosalido, Jessica Binetti, Carmen Aguilar, Javier Ugarte Chicote, Helena Bartra, Laura Artigas, Mireia Coma, Cristóbal Richart, Teresa Auguet

**Affiliations:** 1Grup de Recerca GEMMAIR (AGAUR)—Medicina Aplicada (URV), Departament de Medicina i Cirurgia, Institut d’Investigació Sanitària Pere Virgili (IISPV), Universitat Rovira i Virgili (URV), 43007 Tarragona, Spain; laia.bertran@urv.cat (L.B.); ailende.eigbefoh-addeh@urv.cat (A.E.-A.); marta.portillo.carrasquer@gmail.com (M.P.-C.); andreitabarri18@gmail.com (A.B.-R.); jessica.binetti@gmail.com (J.B.); caguilar.hj23.ics@gencat.cat (C.A.); ugartecj@gmail.com (J.U.C.); cristobalmanuel.richart@urv.cat (C.R.); 2Anaxomics Biotech S.L., 08007 Barcelona, Spain; helena.bartra@anaxomics.com (H.B.); laura.artigas@anaxomics.com (L.A.); mcoma@anaxomics.com (M.C.)

**Keywords:** RUNX1, NAFLD, NASH, metabolism, systems biology

## Abstract

Nonalcoholic fatty liver disease (NAFLD) is the most prevalent chronic hepatic disease; nevertheless, no definitive diagnostic method exists yet, apart from invasive liver biopsy, and nor is there a specific approved treatment. Runt-related transcription factor 1 (RUNX1) plays a major role in angiogenesis and inflammation; however, its link with NAFLD is unclear as controversial results have been reported. Thus, the objective of this work was to determine the proteins involved in the molecular mechanisms between RUNX1 and NAFLD, by means of systems biology. First, a mathematical model that simulates NAFLD pathophysiology was generated by analyzing Anaxomics databases and reviewing available scientific literature. Artificial neural networks established NAFLD pathophysiological processes functionally related to RUNX1: hepatic insulin resistance, lipotoxicity, and hepatic injury-liver fibrosis. Our study indicated that RUNX1 might have a high relationship with hepatic injury-liver fibrosis, and a medium relationship with lipotoxicity and insulin resistance motives. Additionally, we found five RUNX1-regulated proteins with a direct involvement in NAFLD motives, which were NFκB1, NFκB2, TNF, ADIPOQ, and IL-6. In conclusion, we suggested a relationship between RUNX1 and NAFLD since RUNX1 seems to regulate NAFLD molecular pathways, posing it as a potential therapeutic target of NAFLD, although more studies in this field are needed.

## 1. Introduction

Nonalcoholic fatty liver disease (NAFLD) is a condition characterized by excess fat in the liver, without alcohol implication in the onset of the disease. The term NAFLD comprehends a substantial number of liver conditions, ranging from simple steatosis (SS) to the more aggressive form of nonalcoholic steatohepatitis (NASH), which may lead to cirrhosis and hepatocellular carcinoma [1]. SS is defined as the presence of ≥5% hepatic steatosis without evidence of hepatocellular injury in the form of hepatocyte ballooning, inflammation [2], and fibrosis, three remarkable events in NASH pathology. The progress of the disease may vary from individuals, depending on the accumulated fat to the immunological and the oxidant stress responses [3,4].

NAFLD is the most prevalent chronic liver disease, with a global prevalence in adults between 23–25% [5,6]. Nevertheless, nowadays there is no definitive diagnostic test apart from invasive liver biopsy, and no specific approved treatment besides exercise and dietary interventions. Pharmacologic-based therapies for NAFLD are limited, but many clinical trials are in process [7]. For this reason, knowledge about NAFLD pathophysiology is continuously growing.

RUNX1 belongs to the runt-related transcription factor (RUNX) family of genes, and is also known as acute myeloid leukemia 1 [8]. RUNX1 regulates the differentiation of hematopoietic stem cells into mature blood cells [9,10]. It also plays a major role in the development of the neurons that transmit pain [11], and in angiogenesis and inflammation [12]. In addition, RUNX1 involvement in apoptotic processes has been reported on one hand to induce apoptosis and inhibit tumor progression in neuroblastoma [13] and leukaemia [14], while it contrarily seems to present an antiapoptotic effect in pancreatic and ovarian cancer [15,16].

Diseases associated with RUNX1 include platelet disorders with associated myeloid malignancy and blood platelet disease [17]. Related pathways include transport of glucose and other sugars, bile salts, organic acids, metal ions, and amine compounds, as well as transforming growth factor-beta (TGF-β) signaling pathways [18,19]. Recently, Kaur et al. reported a relationship between RUNX1 and NAFLD. Authors related its activity with the progression to NASH, since the interaction of RUNX1 and C-C motif chemokine 2 (CCL2), an important adhesion molecule, mediates the infiltration of pro-inflammatory and pro-angiogenic factors in NASH [20]. Thus, we previously wanted to study the role of RUNX1 mRNA and protein expression in NAFLD in a cohort of women with morbid obesity. We hypothesized that RUNX1 may play a protective role in NAFLD since its expression was enhanced in early stages of the disease and decrease along with the progression to NASH [21]. Given these controversies among our previous results and what was already known, the objective of the present work is to determine the proteins and the potential molecular mechanisms that could establish a link between the activity of RUNX1 and NAFLD pathogenesis by means of systems biology.

## 2. Materials and Methods

### 2.1. Bibliographic and Metadata Analysis in Databases

First, we built the molecular description of NAFLD pathophysiology through systematic searches and reviewing the most up-to-date scientific knowledge regarding this pathology (Appendix A). Accordingly, NAFLD was divided in specific pathophysiological processes–called motives–involved in SS, in NASH, or in both forms (Appendix A), and the corresponding molecular effectors (or key proteins) playing biological roles in these mechanisms were identified (Appendix A).

The interactome around RUNX1 was manually curated in order to better fit the mathematical models. Protein relationship databases including TRRUST database (Transcriptional Regulatory Relationships Unravelled by Sentence-based Text-mining) [22], BioGRID (The Biological General Repository for Interaction Datasets) [23], HPRD (Human Protein Reference Database) [24,25], INTACT (IntAct Molecular Interaction Database) [26], KEGG (Kyoto Encyclopedia of Genes and Genomes) [27], REACTOME (Reactome Pathway Database) [28], and available scientific literature were the sources used to identify and curate new direct interactors of RUNX1.

### 2.2. Mechanistic Model Generation

The compiled information was used to generate a mathematical model that simulate NAFLD pathophysiology by applying Therapeutic Performance Mapping System (TPMS) technology [29], which integrates all available biological, pharmacological, and medical knowledge to simulate human physiology in silico (Appendix A). Then, we used an artificial neural networks (ANNs) strategy [30,31] to analyze these models in order to establish the functional relationships between RUNX1 and NAFLD, considering the motives both together and individually. ANNs evaluate the relationship among protein sets or regions inside the Anaxomics network, providing a predictive score that quantifies the probability of the existence of a functional relationship between the evaluated regions. Each score is associated with a *p*-value that describes the probability of the result being a true positive. The ranking score has been divided into five categories: very high (ANN score > 92; *p* < 0.01), high (ANN score = 78–92; *p* = 0.01–0.05), medium-high (ANN score 71–78; *p* = 0.05–0.1), medium (ANN score 37–71; *p* = 0.1–0.25), low (ANN score < 37; >0.25).

Sampling methods-based mathematical models were then generated to determine the potential molecular mechanisms that could justify our hypothesis:Activation of RUNX1 promoting insulin resistance (IR).Activation of RUNX1 promoting lipotoxicity and hepatic injury and liver fibrosis.

TPMS sampling-based methods trace the most probable mechanisms of action (MoA) or paths, both in biological and mathematical terms, which lead from a stimulus (e.g., activation of RUNX1) to a response (e.g., activation of IR) through the biological human protein network. In this way, it identifies the set of possible MoA that achieve a response when the system is stimulated with the specific stimulus. A population of possible solutions was obtained, and this variability was exploited and analysed to obtain a representation with the most represented paths among the set of possible solutions. A detailed description of the applied methodology was described elsewhere [29,32] and in Appendix B.

## 3. Results

### 3.1. Functional Relationship between RUNX1 and NAFLD: ANNs Analysis

The possible functional relationship between RUNX1 and NAFLD, defined as the set of proteins included in its molecular characterization, has been evaluated by means of ANNs analysis. To deepen our insights, the analysis has also been performed individually for each pathophysiological motive included in NAFLD characterization: (1) increased body fat, (2) hepatic IR, (3) altered fatty acid metabolism, (4) lipotoxicity, and (5) hepatic injury and liver fibrosis. The first three pathophysiological processes occur in both SS and NASH, while the last two only happen in NASH pathophysiology or participate in the progression of NAFLD to NASH.

In this study, the relationship between RUNX1 and NAFLD or individual NAFLD motives has been evaluated, assuming that a possible functional relationship could indicate a participation of RUNX1 in NAFLD pathophysiology, either in promoting or reverting the process, since ANNs only indicate the existence of a possible relationship but not its direction. As shown in Table 1, the results obtained suggest a medium relationship of RUNX1 with the global NAFLD, considering all motives simultaneously.

When considering the motives separately, however, RUNX1 seems to show a high relationship with hepatic injury and liver fibrosis, and a medium relationship with both lipotoxicity and hepatic IR.

The different columns show the ANNs score obtained for NAFLD globally and for each individual pathophysiological motive, some involved in SS and NASH stages, while others are only implicated in NASH. Category splitting was based on *p*-value breaks. RUNX1, runt-related transcription factor 1; NAFLD, nonalcoholic fatty liver disease; SS, simple steatosis; NASH, nonalcoholic steatohepatitis. A darker color indicates a higher ANN score.

The MoA of RUNX1 has been built specifically with regards to the pathophysiological motives–hepatic IR, lipotoxicity and hepatic injury & liver fibrosis–due to their high probability of relationship with RUNX1 and the previously known molecular information found in available scientific literature. Figure 1 shows the protein network of direct RUNX1 interactions with NAFLD effector proteins (the activity of which play a known role in the condition).

### 3.2. Mechanisms of Action of RUNX1

Then, TPMS sampling methods-based mathematical models were generated simulating NAFLD pathophysiology to identify the key proteins and the most probable paths that link the activation of RUNX1 with the most strongly-related motives according to ANNs analysis (hepatic IR, and lipotoxicity, hepatic injury, and liver fibrosis). To provide new insights on the different disease stages (SS or NASH), we studied two independent MoA, considering whether the motive occurs in early or later stages of the disease: (1) RUNX1 promoting IR and (2) RUNX1 promoting lipotoxicity and hepatic injury and fibrosis, respectively.

#### 3.2.1. Mechanism of Action of RUNX1 Promoting IR

Figure 2 summarizes some of the most interesting pathways that could be regulated by RUNX1 in the context of promotion of IR in NAFLD, including the modulation of genes such as CCAAT/enhancer-binding protein alpha (CEBPA), histone deacetylase 1 (HDAC1), the transcription factor c-JUN, nuclear factor kappa B (NFκB), and some types of protein kinase C (PKCβ and PKCε).

Table 2 shows the IR effector proteins that are regulated by the activation of RUNX1 to promote this motive (considering that the activity values of the proteins in our models range from 1 to -1, only proteins with activation state > 0.1 are shown); the table contains all modulated proteins, not only those highlighted by the most represented paths. RUNX1 could be promoting IR through the regulation of 64.10% of the effector proteins involved in this motive. The IR effector proteins most activated by RUNX1-dependent downstream pathways are, in decreasing order: NFκB, JNK, PKCε, tumour necrosis factor (TNF), inhibitor of nuclear factor kappa B kinase subunit beta (IKBKB), and prostaglandin G/H synthase 2 (PTGS2); while the most inhibited ones are insulin receptor substrate (IRS)-1, phosphatase and tensin homolog (PTEN), IRS2 and sirtuin 1 (SIRT1).

NAFLD, nonalcoholic fatty liver disease; NAFLD, nonalcoholic fatty liver disease; RUNX1, runt-related transcription factor 1; MoA, mechanism of action. Green color indicates a positive interaction between the effector protein and RUNX1, while red color indicates a negative one. A more intense color indicates a higher intensity of activation/inhibition.

#### 3.2.2. Mechanism of Action of RUNX1 Promoting Lipotoxicity and Hepatic Injury-Liver Fibrosis

As shown in Figure 3, most of the molecular pathways that may justify the potential role of RUNX1 promoting lipotoxicity and fibrosis-related processes are shared with those involved in the motive IR.

Table 3 describes the lipotoxicity and fibrosis effector proteins that are regulated by the activation of RUNX1 (considering that the activity values of the proteins in our models range from 1 to −1, only proteins with activation state >0.1 are shown). RUNX1 promotes lipotoxicity and fibrosis by the regulation of 50.88% and 62.07% of the effector proteins involved in these motives, respectively. In total, 17 proteins specific to lipotoxicity, 24 to fibrosis, and 12 involved in both motives are regulated by RUNX1. The proteins most regulated by RUNX1 involved in lipotoxicity-related processes are: JNK1, CEBPB, and IKBKB, and those involved in fibrosis-related processes are mothers against decapentaplegic homolog 3 (SMAD3), angiopoietin-2 (ANGPT2), apoptosis regulator BAX, type-1 angiotensin II receptor (AGTR1), and TGF-β. Effector proteins with a role in both pathophysiological processes most activated by RUNX1 are NFκB, NADPH oxidase (NOX)-1, NOX4, CCL2, and TNF. The proteins most inhibited by RUNX1 are SIRT1 (lipotoxicity) and PTEN (fibrosis). Note that the list of proteins in Table 3 is not limited to those shown in the Figure 3.

NAFLD, nonalcoholic fatty liver disease; RUNX1, runt-related transcription factor 1. Green color indicates a positive interaction between the effector protein and RUNX1, while red color indicates a negative one. A more intense color indicates a higher intensity of activation/inhibition.

### 3.3. Overlap between the Mechanistic Pathways Modulated by RUNX1 Activation in IR and Lipotoxicity & Fibrosis Stimulation

The NAFLD motives that have been studied for the generation of the two MoA in this project seem to be pathophysiologically related to each other since there is an overlap of effector proteins from the three motives, as described in Figure 4.

This high relationship prompted us to explore whether an overlap existed in the pathways regulated by the activation of RUNX1 in promoting these motives, and therefore, to be able to relate them. Thus, we have evaluated the similarities that interrelate the motives at the level of common effector proteins and/or pathways modulated by RUNX1 according to our models.

Common NAFLD effector proteins regulated by RUNX1 downstream mechanisms have been recognized by studying the overlap for the three motives together and studying pairs of motives separately. The proteins that we consider to be RUNX1-regulated with an activation value >0.1 are shown in Table 4. In this sense, cannabinoid receptor 1 (CNR1), which was found to be one of the common effector proteins with the three NAFLD analysed motives, presented an activation value lower than 0.1, and it is for this reason that we stop taking this protein into account from now on.

Values of protein activity in each MoA are displayed. “Causative effect in NAFLD” indicates whether the protein is increased/overactivated (1) or reduced/inhibited (−1) in NAFLD. NAFLD, nonalcoholic fatty liver disease; MoA, mechanism of action; IR, insulin resistance; L&F, lipotoxicity and fibrosis; RUNX1, runt-related transcription factor 1; NFκB, nuclear factor kappa B; TNF, tumour necrosis factor; IL6, interleukin 6; ADIPOQ, adiponectin; NOX, NADPH oxidase; CCL2, C-C motif chemokine 2; CYBB, cytochrome b-245 heavy chain; TLR, toll-like receptor; JNK1, c-Jun N-terminal kinase; IKBKB, inhibitor of nuclear factor kappa B subunit beta; MTOR, mammalian target of rapamycin serine/threonine-protein kinase; LCN2, neutrophil gelatinase-associated lipocalin 2; SIRT1, sirtuin 1; PTEN, phosphatase and tensin homolog. Protein codes were obtained from UniProt database.

As shown in Table 4, overlapping of RUNX1-regulated proteins is observed in all three motives and in each pair. Despite finding six effector proteins that share the three NAFLD motives, only five presented sufficient signal intensity to be considered downstream effector proteins of RUNX1 inducing NAFLD; these are NFκB1, NFκB2, TNF, ADIPOQ, and IL-6.

## 4. Discussion

The novelty of this work lies in the fact that we aimed to perform a high-throughput screening to determine the molecular mechanisms that could establish a link between the activity of RUNX1 and NAFLD pathogenesis.

Until now, the connection between RUNX1 and NAFLD remains uncertain. On one hand, Kaur et al. showed a significant correlation between RUNX1 expression and inflammation, fibrosis, and NASH activity score in patients presenting NASH; they also reported RUNX1 function as a pro-angiogenic factor in SS and NASH [20]. On the other hand, Liu et al. presented low levels of RUNX1 in hepatocellular carcinoma. In this sense, these authors suggested that RUNX1 is a tumour suppressing factor that inhibits angiogenesis [33]. Regarding our previous study, we reported that the mRNA and protein expression of RUNX1 in liver seems to be involved in first steps of NAFLD with a proangiogenic-repairing role; meanwhile, RUNX1 appears to be downregulated in the NASH stage [21]. Since these disagreements, an exhaustive study of the relationship between RUNX1 MoA and NAFLD/NASH pathogenesis need to be performed to clarify this issue. In addition, this study could help to recognize RUNX1 as a potential therapeutic target of NAFLD. Previous reports have suggested that RUNX1 could be a potential therapeutic target of cancers, such as acute myeloid leukaemia, since this protein is an important regulator of haematopoiesis in vertebrates [34,35]. The beneficial effect of the therapeutic amelioration of RUNX1 in patients with nonsmall-cell lung cancer has also been described, since the RUNX1 overexpression is correlated with enhanced metastasis [36]. In addition, RUNX1 have been suggested as a potential therapeutic target to limit the progression of adverse cardiac remodeling and heart failure [37,38]. In this regard, to analyze the potential use of RUNX1 as a therapeutic target of NAFLD should be thoroughly studied. For example, investigating liver targeting through liposomes or bile acids in liver cancer [39] could be possible future strategies to evaluate the role of RUNX1 in the pathogenesis of NAFLD.

In this sense, when we performed an ANN analysis concerning the probability of the relationship between RUNX1 protein and NAFLD motives, our first main finding is that RUNX1 seems to show a medium intensity relationship with both motives–hepatic IR and lipotoxicity–and a high intensity relationship with hepatic injury and liver fibrosis motives, suggesting that this protein probably plays a role in these processes. In this regard, this result matches with Kaur et al., who reported a relevant association between RUNX1 expression and fibrosis and inflammation, two of the main NASH parameters [20]. However, this result contradicts our previous reported hypothesis about the potential protective role of RUNX1 in early stages of NAFLD [21]; in contrast, our current result has shown a low or medium intensity relationship with SS-related parameters. Given that our ANN approach provides the probability of functional relationship–regardless of the activity status (up or downregulation)–and the current conflicting results in the literature, further studies in humans or in vivo are required to clarify these contradictions, although the current available evidence clearly supports an involvement, either by presence or absence, of RUNX1 in NAHLD and NASH.

In the current literature, no direct role of RUNX1 on IR has been described yet. However, as a novelty, we demonstrate in the present study that RUNX1 interacts with proteins involved in this pathophysiological process. IR can be defined as a reduced response of the liver to the effects of insulin, which triggers impaired glucose homeostasis (gluconeogenesis and glucose uptake). IR may exert multiple effects on hepatic metabolism such as increased lipogenesis, increased free fatty acids (FFA) uptake, impaired FFA export, and decreased FFA oxidation. Moreover, outside the liver, IR causes increased serum FFA levels because of failure of insulin to suppress hormone sensitive lipase-mediated lipolysis in adipose tissue [3,40,41,42]. In this situation, the PKCε, a downstream intermediate of RUNX1 signaling [20], is activated by the accumulation of diacylglycerol and participates in hepatic IR through impairing insulin signaling [43,44]. In addition, it is believed that RUNX1 could also be involved in IR through the transcription of IL-17 [45], a cytokine that leads to neutrophil and monocyte infiltration in the liver, thereby increasing IR [46]. In contrast, RUNX1 has been shown to inhibit the expression of Suppressor of cytokine signaling (SOCS)-3 [47]—an intracellular protein interfering with insulin signaling via ubiquitin-mediated degradation of IRS1 and IRS2 [48]—therefore ameliorating the IR. According to these facts, RUNX1 seems to have a dual role both promoting and/or preventing hepatic IR.

Another crucial event clearly involved in NAFLD progression is the lipotoxicity resulting from an excessive FFA influx to hepatocytes. Hepatic lipotoxicity occurs when the capacity of the hepatocytes to manage and export FFA as triglycerides is overwhelmed [49]. The molecular mechanisms responsible for lipotoxicity in NAFLD include endoplasmic reticulum and oxidative stress and impaired autophagy, processes that in turn activate apoptotic cascades, thus promoting tissue damage and inflammation [49].

Consequently, in conditions of hypoxia induced by steatosis [50] and inflammation, angiogenesis is triggered in chronic liver diseases [51]. It was demonstrated that proangiogenic factors have an early function in NAFLD progression from SS to NASH since proangiogenic treatments reduce not only inflammation but also steatosis [52]. In this regard, RUNX1, a pivotal regulator of hematopoiesis and angiogenesis [12,53], could be activated in order to repair the liver damage in early stages of NAFLD [21,54]. In contrast, RUNX1 activates target genes involved in lipotoxicity [55,56,57,58] such as CEBPB [59] and Cyclic AMP-dependent transcription factor (AT6) in a regulatory feed-back loop with the transcription factor AP-1 and JNK [60]. If exposure of hepatocytes to lipotoxicity and liver injury continues, it can induce apoptosis [61] and trigger inflammation by interacting with toll-like receptors (TLRs). Inflammation is a component of the wound healing process that leads to fibrosis, the deposition of extracellular matrix in liver parenchyma [62]. Additionally, RUNX1 may contribute to fibrosis and inflammation by modulation of pro-inflammatory cytokines (IL-1β, IL-6, TNF, etc.) [47,63], tissue inhibitor of metalloproteinase 1 (TIMP-1) [64], osteopontin [65] and TLRs [66], among others. Hence, RUNX1 seems to play a dual role, inducing pro-inflammatory cytokines and triggering liver damage, but at the same time having a protective effect by trying to repair the hepatic damage via angiogenesis-related processes.

The second notable finding of this work was obtained because we performed the TPMS technology to identify the key proteins and the most probable paths that link the activation of RUNX1 with the most strongly related motives according to ANNs analysis. In this regard, we wanted to evaluate IR first, since it is one of the main parameters involved in the first stages of NAFLD [67]. Accordingly, IR effector proteins most activated by RUNX1-dependent downstream pathways are NFκB, JNK, PKCε, TNF, IKBKB, and PTGS2, while the most inhibited ones are IRS1, PTEN, IRS2, and SIRT1.

RUNX1 could activate the PTGS2/cyclooxygenase-2 (COX-2) through its interaction with HDAC1 [68,69]. When PTGS2/COX-2 signaling is activated during inflammation in adipose tissue, it can act as a crucial factor for the promotion of obesity-induced IR and fatty liver [70,71]. According to the inflammatory role of PTGS2/COX-2, this enzyme can be induced by growth factors and different cytokines, such as TNF-α, that play a feed-back regulation role [71]. The cytokine TNF-α, produced by adipocytes and macrophages, is highly activated by the downstream mechanisms of RUNX1, particularly via the interaction with the proto-oncogene c-Jun [60,72] or the activation of NFκB [73]. The IκB kinase (IKBK) complex is the master regulator for activation of the NFκB signaling pathway. The kinase complex comprises the two catalytic subunits, IKK1 (IKBKA) and IKK2 (IKBKB), and the regulatory subunit NEMO (IKBKG), which mediates NFκB activation in response to a number of different stimuli such as RUNX1, by phosphorylating IκB proteins [74]. NFκB plays an important role in the regulation of a wide range of proteins/molecular pathways involved in IR. Its activation can be induced by TNF-α and JNK mechanisms [75,76] and can lead to the up-regulation of TNF-α, IL-6 and neutrophil gelatinase-associated lipocalin (LCN-2), contributing to IR-related processes in NAFLD [48,73,77,78]. In addition, the transcription factor AP-1 aggravates IR by inflammation-related processes, inducing the expression of IL-6 [79,80] and TNF-α [72]. TNF-α could be importantly contributing to the development of IR by inhibition and degradation of the IRS mediated by a serine phosphorylation through different mechanisms: (1) SOCS3 is induced by the NFκB/JNK-mediated activation of TNF-α and IL-6 [81], or via CEBPA activation [82], inducing ubiquitin-mediated degradation of IRS1 and IRS2 [83]; (2) MTOR can be activated by TNF-α or PKCβ pathways [84,85] due to hyperglycemia, leading to phosphorylation of multiple serine residues in IRS1 and IRS2 with their further degradation; (3) JNK1 promotes IRS1 and IRS2 serine phosphorylation [86,87]. The inhibitory effects of JNK1 could be also stimulated by PKCβ and PKCε [88,89]. In this regard, some studies have identified associations of PKC activity with disruption of the insulin-induced signal transduction pathway [90,91,92].

In contrast, apart from the degradation/silencing of IRS induced by RUNX1, which was explained above, our analysis has also reported the negative effect of RUNX1 in phosphatase and tensin homologue (PTEN) signaling. Decreased PTEN activity would lead to excessive fat deposition in the liver [40]. PTEN physiological functions negatively regulate the activity of phosphatidylinositol 3-kinase (PI3K)/AKT pathway, which in normal conditions induces lipogenesis in hepatocytes, consequently triggering IR [93,94]. PTEN downregulation has been reported to be carried out by mechanisms involving the sequential activation of MTOR and NFκB [95]. On the other hand, we have also reported a strong repression of SIRT1 by RUNX1 action. SIRT1 is an essential negative regulator of pro-inflammatory pathways, mainly through down-modulating NFκB transcriptional activity, decreasing de novo lipogenesis, and increasing fatty acid β-oxidation [96]. Hence, RUNX1-mediated inhibition of SIRT1 interrupts the beneficial effect of this protein, thus promoting IR. In short, the action of all these effector proteins together gives rise to IR mechanisms.

Regarding the second main finding of this work, we wanted to focus the study of the most implicated motives in NASH stage [97]. In this sense, the effector proteins with a role in NASH (lipotoxicity and fibrosis related processes) most activated by RUNX1 are NFκB, TNF, CCL2, NOX1, and NOX4; the proteins most inhibited by RUNX1 are SIRT1 (lipotoxicity) and PTEN (fibrosis).

NFκB appear to be a relevant regulation core since several RUNX1 interactors regulate its expression [47]. NFκB might be activated by molecular mechanisms such as those explained above (TNF-α/AP-1/JNK pathways). The downstream effects of NFκB activation result in lipotoxicity, hepatocyte injury, inflammation, and fibrosis [40] through upregulated expression of the pro-inflammatory and/or pro-fibrogenic cytokines: CCL2 also called monocyte chemoattractant protein 1 (MCP-1) [98], IL-6 [77] and matrix metalloproteinase-2 (MMP-2) [99]. In particular, higher levels of CCL2 have been identified in NASH subjects in comparison with simple fatty liver [3,100].

Free fatty acids promote hepatic lipotoxicity by stimulating TNF-α expression via a lysosomal pathway, which could be stimulated by the RUNX1/c-Jun link [60,101,102,103,104]. JNK-1, also activated by RUNX1 regulated PKCε activation [20,89], leads to the induction of NFκB dependent pathways [105] and the proapoptotic protein BAX [106], resulting in hepatic tissue damage [107].

Additionally, the isoforms NOX1 and NOX4 seem to be upstream regulated by RUNX1. These proteins show a crucial role on both lipotoxicity and fibrosis-related processes, specially by regulating the activation of hepatic stellate cells and apoptosis, which are two important aspects in the fibrogenic process in NASH [108]. Oxidative biomolecular damage and dysregulated redox signaling induce high oxidative stress and thereby liver injury. Moreover, several studies have shown that the inhibition of NOX1 and NOX4 leads to decreased oxidative stress, lipid peroxidation, hepatic injury, inflammation, and fibrosis in NASH [108,109]. RUNX1 could induce NOX4 expression via PKCε [110,111] and NFκB dependent pathways [112], and induce NOX1 only through PKCε activation [111].

Conversely, RUNX1 have shown to inhibit SIRT1 and PTEN. Some studies have reported that liver-specific disruption of SIRT1 not only causes hepatic steatosis but also promotes the progression to an advanced metabolic disorder stage such as lipotoxicity [113]. Additionally, it seems that dysregulations of PTEN expression/activity in hepatocytes represents an important and recurrent molecular mechanism contributing to the development of liver disorders [114], given that further aberrant activation of hepatic stellate and Kupffer cells trigger the development of liver fibrosis and inflammation [95]. In summary, the pathway that constitute these effector proteins gives rise to processes of lipotoxicity and liver damage.

Accordingly, we have reported for the first-time specific MoA that RUNX1 could play a role in NAFLD pathogenesis motives, but this is only an in silico study and needs to be further validated in experimental research.

The last main objective of the present study is to analyze the overlapping proteins between the studied motives involved in NAFLD. In this sense, the shared proteins between IR and lipotoxicity most activated by RUNX1 are JNK1, IKBKB, MTOR, and LCN2, while the most inhibited by RUNX1 is SIRT1. On the other hand, the overlapping proteins observed in lipotoxicity and fibrosis motives that are the most positively modulated by RUNX1 mechanisms are NADPH oxidase NOX1 and NOX4, the chemokine CCL2, the cytochrome CYBB, and the TLRs 2, 4, and 9. The only effector that is shared between IR and fibrosis negatively modulated by RUNX1 is PTEN. Finally, the main contribution of this study is that we found five RUNX1-regulated proteins with a direct involvement in the three main NAFLD motives, which are NFκB 1, NFκB 2, TNF, ADIPOQ, and IL-6. These proteins are indicators of the relevance of their processes in terms of the relationship with RUNX1 mechanisms towards promoting NAFLD. NFκB1 and TNF present a high activation due to RUNX1 activity, as we explained above. In fact, NFκB-dependent pathways seem to definitely be a key element in these MoA due to its high number of up/downstream links, and for its important regulation of a lot of effector proteins of these motives, especially immune response-related proteins that trigger inflammation, fibrosis, or lipotoxicity [96].

On the other hand, in this study, NFκB 2, IL-6, and ADIPOQ present moderated values of activation, which differ from those of NFκB 1 and TNF. NFκB 2 is an important regulator of RUNX1. It was shown that transcription levels of NFκB 2 were increased in RUNX1-deficient cells [115]. Furthermore, IL-6, as we already mentioned, is a pro-inflammatory cytokine that takes part in fibrosis and tissue damage induced by RUNX1 [47]. High TNF-α and IL-6 levels have been found in NAFLD patients, indicating an important role of these cytokines in the disease. In fact, IL-6 reduction was significantly correlated with both weight loss and insulin sensitivity [48]. Conversely, ADIPOQ seems to be downregulated by RUNX1 signaling. It has been shown that significantly up-regulated ADIPOQ expression in white adipose tissue leads to increased serum adiponectin concentrations. Low adiponectin levels are closely related to the severity of liver histology in NAFLD [116].

Our approach, as all modelling approaches, is subjected to limitations. First, it is limited by the current knowledge on the key studied elements, in this case RUNX1 functions and interactors and NAFLD molecular pathophysiology; thus, the models and conclusions are susceptible to being updated over time if prospective data and new information are generated. Nevertheless, TPMS models are built by considering the whole human protein network and a wide range of drug–pathology relationships (Appendix A); not only limited to the key studied elements, or even to hepatic involvement, they present accuracies against the training set above 80% in the case of ANN models, and above 90% in sampling methods-based models [32]. Thus, systems biology and artificial intelligence approaches allowed us to explore and present mechanistic hypotheses that are in agreement with current knowledge, providing a guide for further pre-clinical investigation in the advancement towards defining treatments for NAFLD. Further studies are needed for confirmation and advancement of these data.

## 5. Conclusions

NAFLD pathophysiological motives most functionally related to RUNX1, according to an ANNs-based analysis, are hepatic insulin resistance, lipotoxicity, and hepatic injury-liver fibrosis. These three pathophysiological processes are molecularly related, since they share NFκB1, NFκB2, TNF, ADIPOQ, and IL-6 as effector proteins. This connection suggested that RUNX1 could regulate molecular pathways involved in NAFLD pathogenesis, but more studies in this field are needed.

## Figures and Tables

**Figure 1 biomedicines-10-01315-f001:**
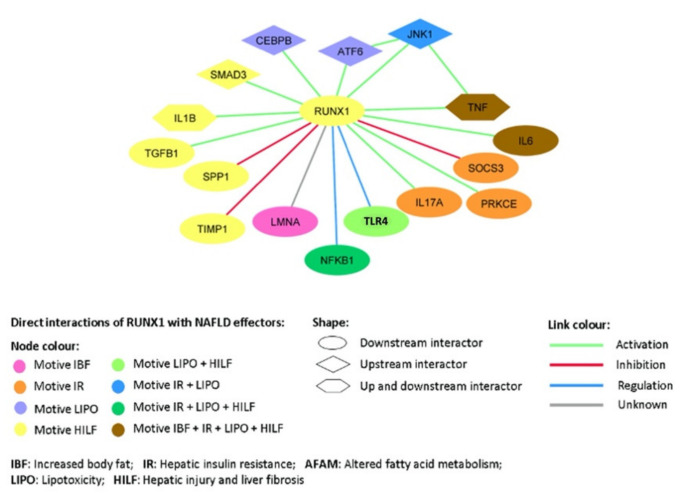
NAFLD effector proteins interacting with RUNX1 at distance 1 (direct link). RUNX1, runt-related protein 1; CEBPB, CCAAT/enhancer-binding protein beta; ATF-6, AMP-dependent transcription factor 6; JNK1, c-Jun N-terminal kinase 1; TNF, tumour necrosis factor; IL6, interleukin 6; SOCS3, suppressor of cytokine signaling 3; PKCE, protein kinase C epsilon type; IL17A, interleukin 17A; TLR4, toll-like receptor 4; NFKB1, nuclear factor kappa B 1; LMNA, lamin-A/C; TIMP1, metalloproteinase inhibitor 1; SPP1, secreted phosphoprotein 1; TFGB1, transforming growth factor beta-1 proprotein; IL1B, interleukin 1 beta; SMAD3, mothers against decapentaplegic homolog 3.

**Figure 2 biomedicines-10-01315-f002:**
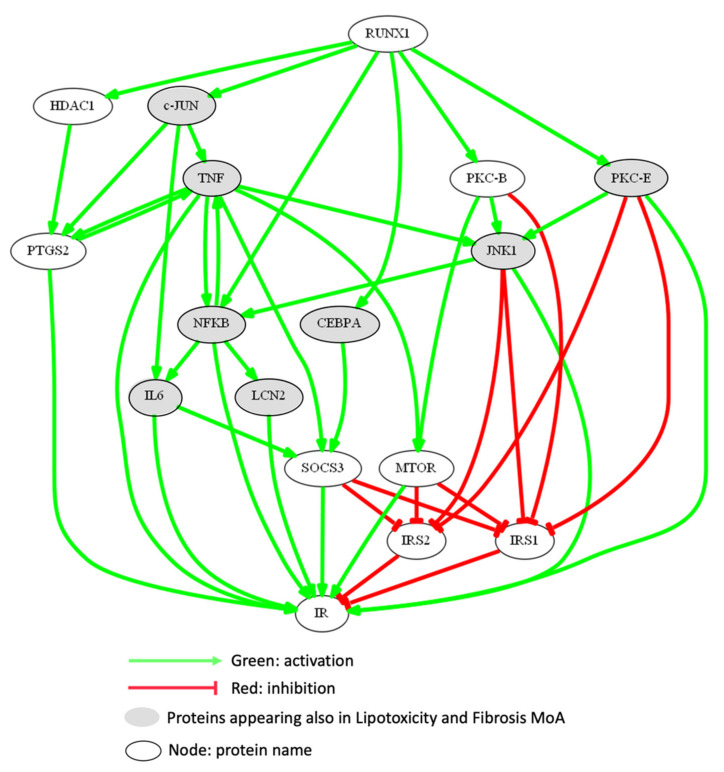
Most represented MoA of RUNX1 promoting IR in NAFLD in the population of TPMS model solutions. Gene names are used in the representations. RUNX1, runt-related protein 1; HDA1C, histone deacetylase 1; PTGS2, prostaglandin G/H synthase 2; c-Jun, protein encoded by JUN gene; TNF, tumour necrosis factor; NFκB, nuclear factor kappa B; IL6, interleukin 6; LCN2, neutrophil gelatinase-associated lipocalin 2; CEBPA, CCAAT/enhancer-binding protein beta; SOCS3, suppressor of cytokine signaling 3; PKC, protein kinase C; JNK1, c-Jun N-terminal kinase 1; MTOR, mammalian target of rapamycin serine/threonine-protein kinase; IRS, insulin receptor substrate. This picture was generated using Graphviz software.

**Figure 3 biomedicines-10-01315-f003:**
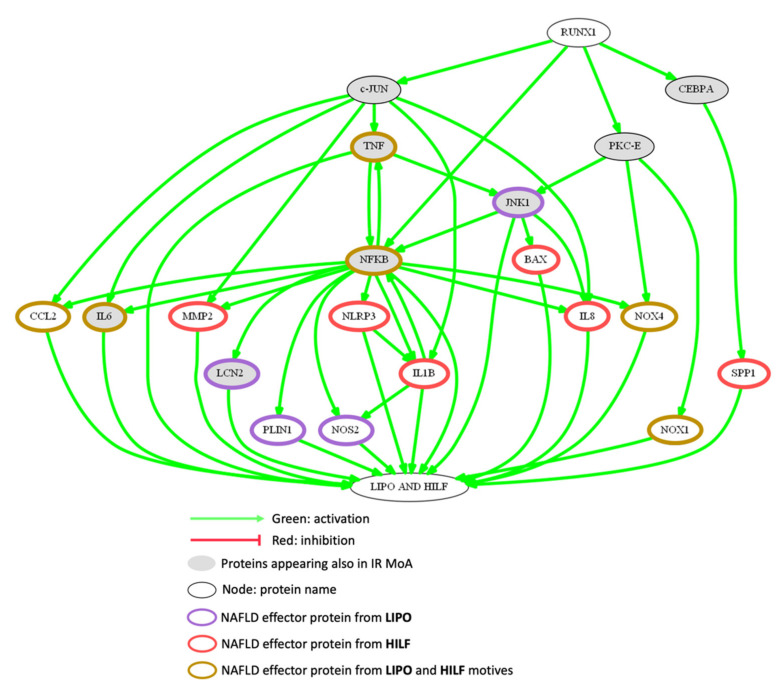
Most represented MoA of RUNX1 promoting lipotoxicity and hepatic injury and fibrosis in NAFLD in the population of TPMS model solutions. Gene names are used in the representations. RUNX1, runt-related protein 1; c-Jun, protein encoded by JUN gene; NFκB, nuclear factor kappa B; PKCε, protein kinase C epsilon; CEBPA, CCAAT/enhancer-binding protein alpha; SPP1, osteopontin; JNK1, c-Jun N-terminal kinase 1; BAX, BCL2 Associated X; CCL2, C-C motif chemokine 2; IL, interleukin; MMP2, matrix metalloproteinase-2; NLRP3, NACHT, LRR and PYD domains-containing protein 3; TLR, toll-like receptor; NOS2, inducible nitric oxide synthase; LCN2, neutrophil gelatinase-associated lipocalin; PLIN1, perlipin; NOX, NADPH oxidase; IR, insulin resistance; LIPO, lipotoxicity; HILF, hepatic injury and liver fibrosis. This picture was generated using Graphviz software.

**Figure 4 biomedicines-10-01315-f004:**
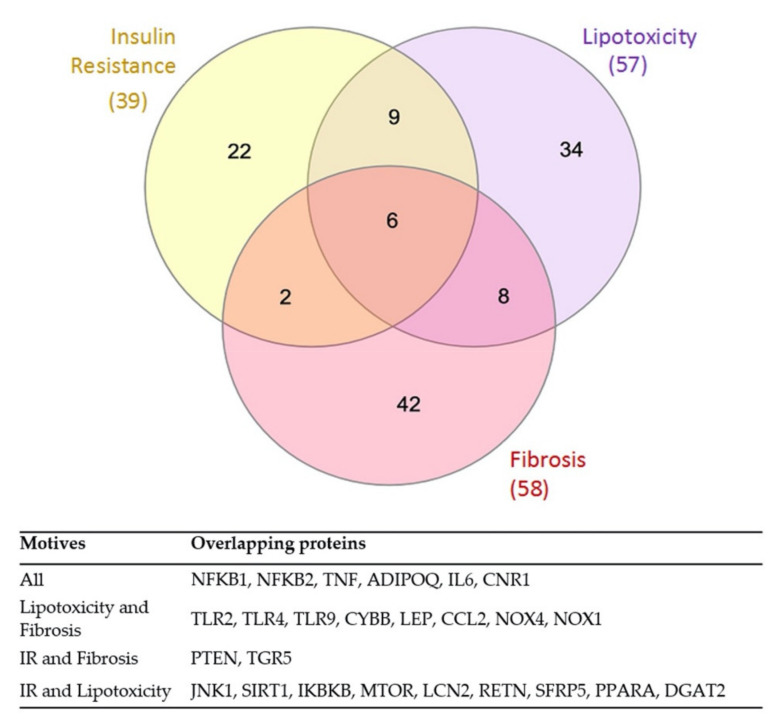
Overlap of effector proteins between the three NAFLD motives evaluated in the project: VENN diagram showing the number of effector proteins overlapping between the three indicated NAFLD motives. There are 39 proteins involved in IR mechanism, 57 in lipotoxicity, and 58 in fibrosis. Concretely, there are 9 proteins involved in IR and lipotoxicity, 6 in lipotoxicity and fibrosis, and only 2 in IR and fibrosis. In this regard, there are six proteins involved in the three motives of NAFLD pathogenesis: NFκB1, NFκB2, TNF, ADIPOQ, IL-6, CNR1. IR, insulin resistance; NFKB, nuclear factor kappa B; TNF, tumour necrosis factor; ADIPOQ, adiponectin; IL6, interleukin 6; CNR1, cannabinoid receptor 1; TLR, toll-like receptor; CYBB, cytochrome b-245 heavy chain; LEP, leptin; CCL2, C-C motif chemokine 2; NOX, NADPH oxidase; PTEN, phosphatase and tensin homolog; TGR5, G protein-coupled bile acid receptor-1; JNK1, c-Jun N-terminal kinase; SIRT1, sirtuin 1; IKBKB, inhibitor of nuclear factor kappa B kinase subunit beta; MTOR, mammalian target of rapamycin serine/threonine-protein kinase; LCN2, neutrophil gelatinase-associated lipocalin 2; RETN, resistin; SFRP5, secreted frizzled-related protein 5; PPARA, peroxisome proliferator-activated receptor alpha; DGAT2, diacylglycerol O-Acyltransferase 2.

**Table 1 biomedicines-10-01315-t001:** ANNs score of the relationship between RUNX1 and NAFLD, both globally and for each NAFLD motive.

	NAFLD	SS/NASH	SS/NASH	SS/NASH	NASH	NASH
		Increased Body Fat	Hepatic Insulin Resistance	Altered Fatty Acid Metabolism	Lipotoxicity	Hepatic Injury and Liver Fibrosis
**RUNX1**	MEDIUM (67%)	LOW (37%)	MEDIUM (67%)	LOW (22%)	MEDIUM (61%)	HIGH (78%)

**Table 2 biomedicines-10-01315-t002:** IR effector proteins modulated by RUNX1 activation. Causative effect indicates whether the protein is increased/overactivated (1) or reduced/inhibited (–1) in NAFLD.

Gene Name	Protein Name	Causative Effect in NAFLD	MoA Activation by RUNX1
NFKB1	Nuclear factor NF-kappa-B p105 subunit	1	1.000
JNK1	c-Jun N-terminal kinase 1	1	0.992
PKC-E	Protein kinase C epsilon type	1	0.883
TNF	Tumor necrosis factor	1	0.875
IKBKB	Inhibitor of nuclear factor kB kinase subunit beta	1	0.859
PTGS2	Prostaglandin G/H synthase 2	1	0.842
IL17A	Interleukin 17A	1	0.688
MTOR	Serine/threonine-protein kinase mTOR	1	0.684
APOC3	Apolipoprotein C-III	1	0.605
NFKB2	Nuclear factor NF-kappa-B p100 subunit	1	0.543
LCN2	Neutrophil gelatinase-associated lipocalin	1	0.457
SOCS3	Suppressor of cytokine signaling 3	1	0.436
INS	Insulin	1	0.422
NT	Neurotensin	1	0.362
IL6	Interleukin-6	1	0.230
CNR1	Cannabinoid receptor 1	1	0.102
ADIPOQ	Adiponectin	−1	−0.184
NRG4	Pro-neuregulin-4, membrane-bound isoform	−1	−0.305
AKT2	RAC-beta serine/threonine-protein kinase	−1	−0.375
PTPN1	Tyrosine-protein phosphatase non-receptor type 1	−1	−0.436
GSK3	Glycogen synthase kinase-3 alpha	−1	−0.504
SIRT1	Sirtuin 1	−1	−0.868
IRS2	Insulin receptor substrate 2	−1	−0.916
PTEN	Phosphatase and tensin homolog	−1	−0.930
IRS1	Insulin receptor substrate 1	−1	−0.996

**Table 3 biomedicines-10-01315-t003:** Lipotoxicity and fibrosis effector proteins modulated when RUNX1 is activated.

Gene Name	Protein Name	Causative Effect in NAFLD	Activation by RUNX1
LIPOTOXICITY			
JNK1	c-Jun N-terminal kinase 1	1	0.999
CEBPB	CCAAT/enhancer-binding protein beta	1	0.825
IKBKB	Inhibitor of nuclear factor kappa-B kinase subunit beta	1	0.819
MAP3K7	Transforming growth factor beta-activated kinase 1/Mitogen-activated protein kinase 7	1	0.722
NOS2	Nitric oxide synthase, inducible	1	0.667
MTOR	Serine/threonine-protein kinase mTOR	1	0.617
LCN2	Neutrophil gelatinase-associated lipocalin	1	0.609
PLIN1	Perilipin-1	1	0.563
HMOX1	Heme oxygenase 1	1	0.490
MAP3K5	Apoptosis signal-regulating kinase 1/mitogen-activated protein kinase 5	1	0.460
PPARG	Peroxisome proliferator-activated receptor gamma	1	0.402
XBP1	X-box-binding protein 1	1	0.327
UCP2	Mitochondrial uncoupling protein 2	1	0.183
ACC1	Acetyl-CoA carboxylase 1	1	0.113
ADIPOR2	Adiponectin receptor protein 2	−1	−0.600
ADIPOR1	Adiponectin receptor protein 1	−1	−0.633
SIRT1	Sirtuin 1	−1	−0.780
FIBROSIS			
SMAD3	Mothers against decapentaplegic homolog 3	1	0.859
ANGPT2	Angiopoietin-2	1	0.854
BAX	Apoptosis regulator BAX	1	0.839
AGTR1	Type-1 angiotensin II receptor	1	0.839
TGFB1	Transforming growth factor beta-1	1	0.827
IL1B	Interleukin-1 beta	1	0.693
IL8	Interleukin-8	1	0.683
MMP9	Matrix metalloproteinase-9	1	0.614
FAS	Tumor necrosis factor receptor superfamily member 6	1	0.518
MMP2	72 kDa type IV collagenase	1	0.508
CTGF	Connective tissue growth factor	1	0.497
CASP1	Caspase-1	1	0.472
NLRP3	NACHT, LRR and PYD domains-containing protein 3	1	0.352
AGT	Angiotensinogen	1	0.340
SPP1	Osteopontin	1	0.330
TIMP1	Metalloproteinase inhibitor 1	1	0.282
MYD88	Myeloid differentiation primary response protein MyD88	1	0.277
PDGFA	Platelet-derived growth factor subunit A	1	0.248
LY96	Lymphocyte antigen 96	1	0.224
COL1A1	Collagen alpha-1(I) chain	1	0.209
COL1A2	Collagen alpha-2(I) chain	1	0.124
NR1H4	Bile acid receptor	−1	−0.441
PTEN	Phosphatase and tensin homolog	−1	−0.958
LIPOTOXICITY AND FIBROSIS			
NFKB1	Nuclear factor NF-kappa-B p105 subunit	1	0.999
NOX1	NADPH oxidase 1	1	0.894
NOX4	NADPH oxidase 4	1	0.822
CCL2	C-C motif chemokine 2	1	0.813
TNF	Tumor necrosis factor	1	0.812
CYBB	Cytochrome b-245 heavy chain	1	0.640
NFKB2	Nuclear factor NF-kappa-B p100 subunit	1	0.535
TLR4	Toll-like receptor 4	1	0.501
IL6	Interleukin-6	1	0.342
TLR2	Toll-like receptor 2	1	0.292
TLR9	Toll-like receptor 9	1	0.200
ADIPOQ	Adiponectin	−1	−0.142

**Table 4 biomedicines-10-01315-t004:** Effector proteins modulated by RUNX1 activation shared by the three motives: lipotoxicity and fibrosis; IR and lipotoxicity; and IR and fibrosis.

Gene Name	Protein Code	Causative Effect in NAFLD	Activity in IR MoA	Activity in L&F MoA	Present in the Most Represented MoA	
					IR (Figure 2)	L&F (Figure 3)
**Common RUNX1-modulated effector proteins in three motives**						
NFKB1	P19838	1	1.000	0.999	Yes	Yes
TNF	P01375	1	0.875	0.812	Yes	Yes
NFKB2	Q00653	1	0.543	0.535	-	-
IL6	P05231	1	0.230	0.342	Yes	Yes
ADIPOQ	Q15848	−1	−0.184	−0.142	-	-
**Common RUNX1-modulated effector proteins in lipotoxicity and fibrosis**						
NOX1	Q9Y5S8	1	-	0.894	-	Yes
NOX4	Q9NPH5	1	-	0.822	-	Yes
CCL2	P13500	1	-	0.813	-	Yes
CYBB	P04839	1	-	0.640	-	-
TLR4	O00206	1	-	0.501	-	-
TLR2	O60603	1	-	0.292	-	-
TLR9	Q9NR96	1	-	0.200	-	-
**Common RUNX1-modulated effector proteins in IR and lipotoxicity**						
JNK1	P45983	1	0.992	0.999	Yes	Yes
IKBKB	O14920	1	0.859	0.819	-	-
MTOR	P42345	1	0.684	0.617	Yes	-
LCN2	P80188	1	0.457	0.609	Yes	Yes
SIRT1	Q96EB6	−1	−0.868	−0.780	-	-
**Common RUNX1-modulated effector proteins in IR and fibrosis**						
PTEN	P60484	−1	−0.930	−0.958	-	-

## Data Availability

Not applicable.

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
