# Peer review of "Identification of the Potential Molecular Mechanisms Linking RUNX1 Activity with Nonalcoholic Fatty Liver Disease, by Means of Systems Biology"

_biomedicines, 2022, doi:10.3390/biomedicines10061315_

Round 1

Reviewer 2 Report

In this report, the authors conduct an analysis of the association between the transcription factor RUNX1 and non-alcoholic fatty liver disease (NAFLD). The review of available scientific literature and the generation of a mathematical model using Anaxomics databases demonstrated a connection between RUNX1 and NAFLD as well as other RUNX1-regulated proteins.

Minor points

  1. In abstract they state that they found 6 RUNX1-regulated proteins but only list 5.
  2. I would add a reference to the statement that RUNX1 is also known as acute myeloid leukemia 1.
  3. Throughout the paper there is a poor use of capitalization. For instance, on page 4, lipotoxicity, hepatic, histone, etc. are capatilized when they should not be.
  4. page 4; spell out MoA (mechanism of action) before using the acronym.
  5.  In Figure 4, CNR1 is listed as a protein involved in all motives, but it is not listed in Table 4. Why the discrepancy?  Please check for overall consistency between this figure and table for other proteins.
  6. This inconsistency between mentioning six proteins and then only listing five is throughout the manuscript. It appears that it is always CNR1 that is omitted.  Please correct or clarify
  7. I would suggest that the authors shorten the conclusions a bit. They are long and wordy.

Reviewer 3 Report

Manuscript seems written well. However, this looks review article.

To be an original article, authors should add own patients’ data.

  1. Table 1 is not clear.
  2. Authors should mention about the association between RUNX1 and apoptosis. See the following reference: Kanda T, et al. Apoptosis and non-alcoholic fatty liver diseases. World J Gastroenterol. 2018 Jul 7;24(25):2661-2672. doi: 10.3748/wjg.v24.i25.2661. PMID: 29991872

Round 2

Reviewer 1 Report

The authors have partially addressed my comments, however, given the conflicting data/evidences for RUNX1 in NAFLD and NASH further studies are required to draw a conclusion on its function. In silico models are only valid if they are experimentally tested and verified and this is missing in this study.

Author Response

Thank you so much to contribute in this review process, your comments and suggestions did improve our manuscript. Regarding your first comment, we agree with you, there is really necessary further research in this topic to define a concrete conclusion about the function of RUNX1 in NAFLD/NASH. In this regard, we are very interested in further investigating the role of RUNX1 in NAFLD in an experimental study. Regarding your last comment, we believe it is important to mention that this is an in silico study, which obviously uses bibliographic data, but also mathematical models and high-throughput screening technologies to perform a simulation of molecular mechanisms. Hence, a methodology has been carried out and databases and computational models have been used to obtain our results. The same happens with systematic reviews and meta-analyses that, although they use already created evidence, they apply a methodology to obtain standardized conclusions and are considered "original articles" even though they have not generated "own data". In this sense, it is true that the study could have an experimental validation, but at this time, we do not have the necessary resources to perform it, and given the interesting results obtained, as we alteady mentioned, we have in mind to design an experimental study with cell cultures and animal models to validate the discovered effector proteins induced by RUNX1 that promote the progression of NAFLD. In addition, we would like to mention that we have already carried out an analysis of the role of RUNX1 in the liver of women with morbid obesity and NAFLD, already cited in this work (Bertran L., et al. The Potential Protective Role of RUNX1 in Nonalcoholic Fatty Liver Disease. International Journal of Molecular Sciences 2021). In this study, we build TPMS models in collaboration with Anaxomics (the company that developed this technology) including transcriptomics data generated by our group, to build mathematical models centred on the area of study. Therefore, we want to mention some recent articles that Anaxomics' authors have published in the filed of systems biology without experimental validation:

  • Mechanisms of action of sacubitril/valsartan on cardiac remodeling: a systems biology approach. NPJ Syst Biol Appl, 2017
  • In-silico drug repurposing study predicts the combination of pirfenidone and melatonin as a promising candidate therapy to reduce SARS-CoV-2 infection progression and respiratory distress caused by cytokine storm. PLoS One, 2020
  • Elucidating the Mechanism of Action of the Attributed Immunomodulatory Role of Eltrombopag in Primary Immune Thrombocytopenia: An in Silico Approach. Int J Mol Sci, 2021
  • Proteomics based drug repositioning applied to improve in vitro fertilization implantation: an artificial intelligence model. Syst Biol Reprod Med, 2021
  • Head to head evaluation of second generation ALK inhibitors brigatinib and alectinib as first-line treatment for ALK+ NSCLC using an in silico systems biology-based approach. Oncotarget, 2021
  • Decoding empagliflozin's molecular mechanism of action in heart failure with preserved ejection fraction using artificial intelligence. Sci Rep, 2021

Reviewer 3 Report

All queries have been addressed.

Author Response

Thank you very much for being part of this review process. Your comments and suggestions did improved our manuscript a lot.